

# Impact of the 2011 off the Pacific coast of Tohoku earthquake on a deep-sea benthic ecosystem: evidence from living and dead benthic foraminifera on the landward slope of the Japan Trench

Akira Tsujimoto[1], Ritsuo Nomura[1], Hidetaka Nomaki[2], Kazuno Arai[3], Mutsuo Inoue[4], and Katsunori Fujikura[2]

[1]Faculty of Education, Shimane University, 1060 Nishikawatsucho, Matsue, Shimane 690-8504, Japan

[2]Japan Agency for Marine-Earth Science and Technology (JAMSTEC), 2-15 Natsushima-cho, Yokosuka, Kanagawa 237-0061, Japan

[3]Center for Advanced Marine Core Research, Kochi University, B200 Monobe, Nankoku, Kochi 783-8502, Japan

[4]Low Level Radioactivity Laboratory, Kanazawa University, Wake, Nomi, Ishikawa 923-1224, Japan

*Correspondence to:* A. Tsujimoto (tsujimoto@edu.shimane-u.ac.jp)

**Abstract.** We examined the impact of the earthquake and tsunami following the 2011 off the Pacific coast of Tohoku earthquake on the deep-sea benthic ecosystems based on radionuclide and benthic foraminiferal analysis of core sediments, collected from 3200 and 3600 m water depths 5 and 17 months after the earthquake. Radionuclide analysis of the excess $^{210}$Pb, $^{134}$Cs, and $^{137}$Cs indicated that some of the analyzed sediment core recorded deposits before the earthquake, event deposits just after the

earthquake, and deposits after the Fukushima Daiichi Nuclear Power Plant accident, which caused the release of a large amount of radioactive material 4 days after the earthquake. *Uvigerina senticosa*, *Chilostomella oolina*, and *Elphidium batialis* were the dominant species in the study area prior to the earthquake. In core 4W-2012, the original or pre-earthquake assemblage layer was covered by 5-cm-thick event deposits following the earthquake that contained a high diversity allochthonous foraminiferal assemblage. Following the episodic deposition, foraminiferal density drastically decreased and many species disappeared,

resulting in a decrease in species diversity. Above 10 cm depth in the sediment, living specimens of opportunistic and competitive species gradually increased toward the sediment surface and became dominant in the top 1 cm of the core. Thus, the episodic deposition resulting from the earthquake caused a drastic decrease in the original benthic foraminifera and colonization of opportunistic species with a low diversity within 17 months. Although there were differences in vertical change in the radionuclides and benthic foraminifera between sites, faunal change may have already occurred 5 months after the

earthquake.





## 1. Introduction

The 2011 off the Pacific coast of Tohoku earthquake (hereafter the 2011 Tohoku-oki earthquake, Mw 9.0) occurred on March 11, 2011, off the northeast coast of Japan. The earthquake and subsequent tsunami caused seafloor disturbance in wide-ranging areas of the littoral zone, continental shelfs, continental slopes, and hadal zones (Arai et al., 2013; Oguri et al.,

2013; Ikehara et al., 2014; Tamura et al., 2015). Some studies concerning the effect of the tsunami disturbances on the littoral benthic communities have been conducted (e.g., Seike et al., 2013; Urabe et al., 2013; Kanaya et al., 2015; Miura et al., 2017). These studies indicated that the immediate impacts of the tsunami disturbances on the littoral communities were changes in faunal composition, loss of species diversity, a reduction in the density, and an increase in the density of opportunistic taxa. The tsunami disturbances also affected benthic communities on the continental shelf and upper continental slope (Toyofuku et

al., 2014). Five months after the tsunami, the shelf ecosystems off Shimokita (northeast Japan) were characterized by high species diversity and did not show a reduction in density, but an opportunistic fauna dominated the deeper site (211 m depth). Submarine landslides and tsunami-generated turbidity currents caused mass sedimentation in a wide area of the continental slope and the Japan Trench off Tohoku (Arai et al., 2013; Oguri et al., 2013; Ikehara et al., 2016). The mass sedimentation events must have affected the deep-sea ecosystems as was the case in the littoral ecosystem, but little is known regarding the

impact, in particular deeper than 3000 m (e.g., Oguri et al., 2013; Kitahashi et al., 2014, 2016; Nomaki et al., 2016).

Benthic foraminifera (single-celled protists) are the most common eukaryotes in deep sea benthic communities, and they sometimes account for 50% or more of the eukaryotic biomass (Gooday et al., 1992; Nomaki et al., 2005). Their assemblages are influenced by various environmental factors, such as temperature, carbonate saturation state, oxygen concentration of the bottom water, and the quality and quantity of organic matter reaching the sediment surface (Murray, 2006;

Jorissen et al., 2007 and references therein).

Their calcite or agglutinated tests (shells) are abundantly preserved in sediment cores and can be used to reconstruct environmental changes in benthic communities. They have often been used as indicators of tsunami deposits mostly in shallow water areas based on their bathymetric distribution (Mamo et al., 2009 and references therein). While the shallow-water sediments preserve event deposits generated by both storms and earthquakes, which are difficult to separate (Tamura et al.,

2015), off shore, deep-sea sediments are expected to have event deposits caused by earthquakes or subsequent tsunamis. Usami et al. (2017) studied benthic foraminiferal assemblages within turbidites collected from the upper slope sites off Sendai Bay (NE Japan) after the 2011 Tohoku-oki earthquake and concluded that benthic foraminiferal fauna can be a good indicator of turbidite paleoseismology. For precise reconstructions of paleoseismic events using sediment cores, it is essential to elucidate foraminiferal faunal changes before and after earthquakes, in particular in deep-sea environments.

In this study, we examined the sediment characteristics and foraminiferal assemblages in the sediment cores collected from the landward slope of the Japan Trench around the hypocentral region 5 and 17 months after the earthquake. The event deposits caused by the earthquake and tsunami were determined using sediment granulometry and radionuclide analysis, namely $^{210}$Pb, $^{134}$Cs, and $^{137}$Cs. Temporal changes in the foraminiferal assemblages and their diversities were examined using pre-earthquake, event deposit, and post event sediments. This is the first record documenting the impact of

large-scale seafloor disturbance on a deep-sea (deeper than 3000 m) benthic ecosystem using living and dead benthic foraminifera.

## 2. Materials and methods

### 2.1 Sampling procedure

Sediment core samples were collected using the human-occupied vehicle (HOV) *Shinkai 6500* during the YK11-E06 and the YK12-13 cruises of *R/V Yokosuka* in August 2011 and August 2012 (Fig. 1; Table 1). A push corer with an inner



diameter of 8.2 cm was used to collect surface sediment (9 cm to 20 cm in length) at two different sites (2W: 38°39′N, 143°35′E, 3230 m water depth; 4W: 37°44′N, 143°17′E, ~3570 m water depth) for each cruise; therefore, a total of four cores were examined. Both sites were on a trench slope, but site 4W was along a canyon (Fig. 1). Based on seismic reflection profiles, Tsuji et al. (2013) reported that both sites 2W and 4W are along topographic traces of fault systems; a buried normal fault at

2W and a landward dipping normal fault at 4W. Fissures on the seafloor and bacterial mats associated with mass deposition of megabenthos attributed to the 2011 Tohoku-oki earthquake were observed at 2W and 4W, respectively (Tsuji et al., 2013). Our sediment cores were collected from normal seafloor, where no fissure and bacterial mat was nearby. The site 2W cores were collected ~300 m eastward from the fault scarp with a height of ~150 m (Fig 1B). The site 4W cores were collected at the canyon axes of a small north-southward canyon along the fault (Fig 1C). Exact locations of the push corer sampling were

slightly different between 2011 and 2012 in the 4W area (Fig. 1).

On board, the overlying water was gently removed from the cores using a silicon tube. The sediment cores were subsampled into 0.5-cm-thick slices using a core extruder and then subdivided into two aliquots to separately analyze radionuclide and foraminiferal assemblage. The sliced sediments were stored at 4°C or -80°C for radionuclide and foraminiferal assemblage analyses, respectively.

## 2.2 Water and mud content analyses

In a laboratory on land, the sliced subsamples were oven-dried. Water contents were determined from the ratio of the wet and dry weights of the bulk sediments. Mud contents were determined based on the dry weights of the bulk samples calculated by the water content and the dry weight of the residues of the washed samples (>63-µm).

## 2.3 γ-Spectrometry

The oven-dried subsamples were pulverized to silt size using an agate mortar, after which 2 g of dry-weight sample was sealed in a styrene tube. The samples were left for more than three weeks to allow for radioactive secular equilibrium for $^{222}$Rn and its daughter $^{214}$Pb and $^{214}$Bi with $^{226}$Ra. Low-background γ-spectrometry was performed on the sediment samples

using well-type Ge-detectors (Canberra EGPC150-P16; FWHM resolution 1.4 keV at 122 keV) for one counting day. For the calibration of $^{210}$Pb and $^{214}$Ra, we used the uranium standard issued from the New Brunswick Laboratory, USA (NBL-42-1). The excess of $^{210}$Pb concentration was calculated by subtracting the weighted average of the 242, 295, and 352 keV of $^{214}$Pb concentration from the total $^{210}$Pb concentration (Irizuki et al., 2015).

We also calibrated the $^{134}$Cs and $^{137}$Cs concentrations in the sediment samples by comparing mock-up samples

prepared using a reference material (JSAC0471) based on the γ-ray peaks of $^{134}$Cs at 569 keV and 605 keV, and $^{137}$Cs at 662 keV, respectively. The analytical precision for measuring $^{134}$Cs was 7–38 %, based on the standard deviation of the counting statistics.

All concentration data in the present study were decay-corrected to the sampling date, and the $^{134}$Cs/$^{137}$Cs ratio was decay-corrected to March 15, 2011, when major depositional events of radionuclides on land and the ocean surface occurred

as a result of the Fukushima Daiichi Nuclear Power Plant accident.

## 2.4 Foraminiferal analysis

Sliced subsamples were wet-sieved through a 63-µm sieve. The residues of the depth intervals of the upper 10 cm were stained with Rose Bengal solution (1 g L$^{-1}$) for 24 h to distinguish live from dead individuals (Walton, 1952). The residues

were oven-dried, and then dry-sieved through a 106-µm sieve. Living specimens were identified by bright pink stained cytoplasm of at least few chambers (Schonfeld et al., 2013). Living and dead specimens of the >106-µm fraction were dry-




picked and analyzed. Sediment samples containing abundant foraminiferal specimens were split into fractions containing approximately 200 specimens in total (i.e. living and dead). The foraminiferal density (i.e. number of foraminifera per gram of dry sediment) was calculated from the wet weights of the samples and the water content.

We determined the Shannon Index (H') for the total, dead, and living assemblages using samples containing more than 30 individuals and the PAST software package (version 3.16) (Krebs, 1989; Hammer and others, 2001). A Q-mode cluster analysis was also performed for total (i.e. living and dead) foraminiferal count data using the unweighted pair-group arithmetic average method of the PAST software package. The similarities used were Horn's overlap indices (Horn, 1966).

We performed statistical analysis only for core 4W-2012 because the others did not contain sufficient specimens (>200 individuals per sample).

## 3. Results

### 3.1 Mud and water content

The mud and water contents were nearly stable throughout cores 4W-2011, 2W-2011, and 2W-2012 except for the water contents of the top cm of the cores (Fig. 2). Core 4W-2012 showed characteristic changes in mud and water contents throughout. The mud contents in core 4W-2012 were stable at approximately 90–95% from the bottom of the core (20 cm depth) to a depth of ca. 14 cm, but the values markedly decreased down to 54% and fluctuated between the ca. 14 to 9 cm depths (Fig. 2). The mud contents in core 4W-2012 became stable at approximately 95–100% from ca. 9 cm depth to the top of the core.

The water contents in core 4W-2012 were stable at approximately 70% from the bottom of the core (20 cm depth) to a depth of ca. 14 cm. Between ~14 cm to 9 cm depth, water contents fluctuated considerably, as seen as in the mud contents. The water contents in core 4W-2012 gradually increased from ca. 9 cm depth to the top of the core.

### 3.2 Vertical profiles of radionuclides

The profiles of the excess $^{210}$Pb concentrations are shown in Figure 3. The excess $^{210}$Pb concentrations in core 4W-2011 were quite low throughout, thus we analyzed only the top 4 cm of the core. Here, the concentrations exponentially decreased downward. The excess $^{210}$Pb concentrations in cores 2W-2011 and 2W-2012 were nearly stable throughout the cores. The excess $^{210}$Pb concentrations in core 4W-2012 were low and stable below ca. 15 cm depth, then increased sharply towards 13 cm in depth, and became nearly stable above ca. 9 cm depth. Although $^{134}$Cs was detected only in the upper 1.5 cm in cores 2W-2011 and 2W-2012, $^{134}$Cs was detected in the top 0–10 cm in core 4W-2012, and the two radiocesium isotopes exhibited the same trend (Fig. 4; Table 2). $^{134}$Cs was released into the environment by the Fukushima Daiichi Nuclear Power Plant (FNPP1) accident of March 11, 2011 (Mathieu et al., 2018). Major depositional events of radionuclides on land and the ocean surface occurred in particular from March 14–16 (Mathieu et al., 2018).

### 3.3 Benthic foraminiferal analysis

There were few living foraminifera and the total foraminiferal density decreased upward in core 4W-2011 (Fig. 5). Various species such as *Eilohedra rotunda*, *Elphidium batialis*, and *Pullenia bulloides* nearly disappeared above ca. 5 cm depth in the core, and *Uvigerina senticosa* above ca. 3 cm. Living foraminifera were found throughout the upper 10 cm of core 2W-2011 (Fig. 6). The dominant species were *Nonionella* spp., *Nonionellina labradorica*, *Brizalina pacifica*, and *Stainforthia* cf. *apertura* in order of their abundance (Fig. 6). Living specimens of these species showed the highest density between ca. 1 to 4 cm depth in the core (Fig. 6). Although foraminiferal density was low in core 2W-2012, living specimens of *B. pacifica*, *N. labradorica*, and *Nonionella* spp. showed their highest density between ca. 0 to 2.5 cm depth in the core (Fig. 7). Whereas




cores 4W-2011, 2W-2011, and 2W-2012 contained few foraminiferal tests, core 4W-2012 contained a relatively high number. Density of the dead foraminifera was low (ranging from 14 to 53 individuals per g of dry sediment) from the bottom of the core to ca. 13 cm, but it showed a spike peak (382 individuals g-1) between ca. 12 to 9 cm (Fig. 8). The value decreased to a range from 4 to 12 individuals above ca. 9 cm depth except for the top of the core (Fig. 8). Living foraminifera were found

throughout the upper 10 cm of the core, with the highest density at the sediment surface and a linear decrease down to ca. 5 cm depth in the core. The ratio of living foraminifera to total (living and dead) foraminifera (L/T ratio) increased from 10 cm to the upper part of the core, and reached a maximum (ca. 97%) at approximately 2 cm depth (Fig. 8).

Species diversity (H') was measured to assess the temporal change in foraminiferal biodiversity of core 4W-2012 (Fig. 8). Total diversity showed high values between ca. 14 to 9 cm depth, then decreased to the upper part of the core, and

reached minimum values at approximately 4 cm depth. Living foraminiferal diversity gradually increased from ca. 7 cm depth to the upper part of the core, and reached a maximum at approximately 3 cm depth (Fig. 8).

Two major clusters (I, II) were distinguished for total foraminiferal assemblages based on the Q-mode cluster analysis for core 4W-2012. Vertical distributions of the absolute and relative abundances of the dominant foraminiferal species are shown with Q-mode cluster groupings in figure 9. Cluster I was divided into two sub-clusters (Ia and Ib) by the composition

of the species. Sub-cluster Ia was characterized by the dominance of *Uvigerina senticosa* and *Chilostomella oolina*, with minor *Fursenkoina complanata*, *Brizalina pacifica*, and *Tosaia hanzawai*. Sub-cluster Ib was characterized by the dominance of *Nonionellina labradorica* and *Elphidium batialis*, with minor *F. complanata*, *B. pacifica*, and *T. hanzawai*. These species showed spike peaks between ca. 12 to 9 cm depth (Fig. 9). Cluster II was divided into three subclusters (IIa, IIb, and IIc) based on species composition. The densities were quite low in subcluster IIa, and this subcluster was characterized by the dominance

of *E. batialis*, with minor *B. pacifica* (Fig. 9). Subcluster IIb was characterized by the dominance of *B. pacifica* and *Stainforthia* cf. *apertura*, with minor *E. batialis* and *N. labradorica*. Subcluster IIc was characterized by the dominance of *Stainforthia* cf. *apertura*, *N. labradorica*, and *Nonionella* spp., with minor *B. pacifica*. The species abundant in subcluster IIc were mostly living specimens (Fig. 9).

## 4. Discussion

**4.1 Sea floor disturbance resulting from the 2011 Tohoku-oki Earthquake and the influx of $^{134}$Cs and $^{137}$Cs to the landward slope of the Japan Trench**

$^{210}$Pb has a half-life of 22.3 years and has been used to estimate depositional ages up to a maximum of ca. 150 years, i.e. up to 7 half-lives (Lowe and Walker, 2014). $^{137}$Cs is an artificial radionuclide, and has been detected in sediments from

1954 onwards, with a maximum in 1963 because of intensive releases from aerial nuclear tests (Lowe and Walker, 2014). $^{134}$Cs was released into the environment after the Fukushima Daiichi Nuclear Power Plant (FNPP1) accident, and the first major releases occurred on March 15 (Mathieu et al., 2018). These radionuclides, particularly short-lived $^{134}$Cs (2.06 years), have been used to distinguish post-sediment from pre-sediment of the 2011 Tohoku-oki Earthquake (Oguri et al., 2013; Toyofuku et al., 2014; Ikehara et al., 2014, 2016).

The very low $^{210}$Pb concentrations in core 4W-2011 suggest that the surface sediments at this site were eroded prior to the sampling. Although cores 4W-2011 and 4W-2012 were collected in the same area, slight locational changes in the canyon may result in different sedimentation environments; 4W-2011 was collected from a steeper slope while 4W-2012 was collected from a gentle slope of the same canyon (Fig. 1-C).

$^{210}$Pb concentrations in cores 2W-2011 and 2W-2012 indicate sediment mixing and/or a high sedimentation rate. In

core 4W-2012, the low concentrations of the excess $^{210}$Pb below 15 cm depth indicate that these sediments are sufficiently old. Moreover, the rapid increase in the concentrations of the excess $^{210}$Pb at approximately 15 cm depth indicates that there is a





sedimentation gap at this depth. This is confirmed by the markedly decreased mud contents and fluctuation above ca. 14 cm depth. These results and the fluctuation of the excess $^{210}$Pb between 15 cm to 9 cm, combined with the lack of detectable $^{134}$Cs in this interval, suggest that these sediments facies are event deposits that were deposited immediately after the 2011 Tohoku-oki Earthquake.

5        Because of its short half-life (2.06 y), $^{134}$Cs detected in the sediment samples was derived entirely from the FNPP1 accident. $^{134}$Cs was detected in the top 0–1.5 cm of cores 2W-2011 and 2W-2012, and the top 0–9.5 cm of core 4W-2012 (Fig. 4 and Table 2). The $^{134}$Cs/$^{137}$Cs ratio released from the FNPP1 is known to be approximately 1 (Komori et al., 2013; Kobayashi et al., 2015, 2017). The $^{134}$Cs/$^{137}$Cs ratios in the upper 9 cm depth of core 4W-2012 range from 0.85 to 1.12 (average; 0.95) (Fig. 4 and Table 2). Thus, we confirmed that this interval was deposited following the FNPP1 accident. Honda et al. (2013) analyzed timeseries sinking particles in sediment traps deployed 950 km southeast from the FNPP1 before and after the FNPP1 accident, which caused major deposition of radionuclides on land and the ocean surface from March 14–16. The authors reported that $^{134}$Cs and $^{137}$Cs from the FNPP1 accident were detected in sinking particles collected at 500 m after late March 2011 and at 4810 m after early April 2011. Oguri et al. (2013) detected $^{134}$Cs in a sediment core collected at a water depth of 7261 m in the Japan Trench. The authors suggested that diatom blooms accelerated rapid deposition of $^{134}$Cs in the hadal environment. The Kuroshio-Oyashio Transition zone is situated in the northwest Pacific Ocean, where the mixing of cold nutrient-rich Oyashio waters and warm oligotrophic Kuroshio waters occurs (Inagake and Saitoh, 1998; Itoh and Sugimoto, 2002). In the Kuroshio-Oyashio Transition zone, a spring bloom of primary production occurs because of eddy-driven nutrient fluxes (Inagake and Saitoh, 1998; Sasai et al., 2007; Sasai et al., 2010). The spring bloom is mainly composed of diatoms (Isada et al., 2009; Yatsu et al., 2013), and a large influx of diatoms is deposited on the seafloor (Ikehara et al., 2016). Based on our microscopic observations, although radiolarians were dominant in the >106-μm fraction in the lower part of core 4W-2012 by microscopic qualitative analysis, diatoms became dominant in the upper part (above ca. 9 cm depth). The gradual decrease in water content with increasing sediment depth suggests continuous sedimentation of diatoms in this interval. Thus, we concluded that the sediments of the upper 10 cm of core 4W-2012 were deposited from March 2011 to August 2012 (within 17 months), probably related to the mass bloom of the phytoplankton. Although a sedimentation rate of ~6 cm per year is quite high, similar rapid sedimentation rates have sometimes been reported in depositional environments such as submarine canyons or trenches (Oguri et al., 2013). Cores 2W-2011 and 2W-2012 were collected at the same site, and the interval was approximately 1 year. Detection of $^{134}$Cs in the top 0–1.5 cm of both cores may indicate mass sedimentation did not occur from August 2011 to August 2012 at this site. Differences in vertical change in radionuclides between the sites may indicate that sedimentary and erosional processes were different because of differences in surface productivity or topography, such as the small canyons observed at 4W (Fig 1).

Based on these observations, we divided the sediments of core 4W-2012 into three facies as follows: I) deposits before the earthquake (20–15 cm depth), II) event deposits immediately after the earthquake (15–9 cm depth), and III) deposits after the FNPP1 accident (above 9 cm depth) based on the changes in radionuclides. We discuss the benthic foraminiferal faunal change of the deep-sea floor disturbance focusing on the data of core 4W-2012 because the others do not contain sufficient specimens and are not appropriate for detailed discussion.

### 4.2 Benthic foraminiferal faunal change of the deep-sea floor disturbance

Changes in benthic foraminiferal assemblages of core 4W-2012 coincide with three sediment facies (facies I, II, and III, as previously mentioned). Cluster Ia of the benthic foraminifera is equivalent to sediment facies I, pre-earthquake sediments. The dominant and characteristic species of cluster Ia were *Uvigerina senticosa* and *Chilostomella oolina*. Thompson (1980) reported the bathymetric distributions of dominant benthic foraminiferal species in the Japan Trench area using sediment





samples from Deep Sea Drilling Program Leg56 sites and Lamont Doherty Geological Observatory (LDGO) core collections (Fig. 10). Although the main study area of Thompson (1980) was the northern part of the Japan Trench area (Fig. 1), *U. senticosa* was the dominant species at the approximate water depth of our study site (3585 m) (Fig. 10). *C. oolina* is not documented at an approximate water depth of 3500 m in Thompson (1980), but this species is reported as a characteristic species at depths greater than 3000 m in a different area (Schmiedl and Mackensen, 1997; Uchimura et al., 2017). The most dominant foraminiferal species below 5 cm in core 4W-2011 is *U. senticosa* (Fig. 5), and this interval is sufficiently old based on the $^{210}$Pb profile. Thus, cluster Ia of core 4W-2012 indicates an assemblage before the effects of the earthquake.

Cluster Ib of benthic foraminifera is equivalent to the sediment facies II event deposits deposited immediately following the earthquake. *U. senticosa* and *C. oolina* are the dominant species in cluster Ia, and gradually decrease and nearly disappear at the top of the cluster; *Nonionellina labradorica* and *Elphidium batialis* become dominant towards the top of cluster Ib. Various species appeared in this cluster, resulting in high species diversity (Fig. 8). These species in general are distributed at water depths of 1800–3400 m (Fig. 10; Thompson, 1980). *Bolivina spissa*, nearly only found from cluster Ib, has been reported in the oxygen minimum zone of the Pacific Ocean (Ingle et al., 1980; Quinterno and Gardner, 1987; Fontanier et al., 2014). Fontanier et al. (2014) studied living benthic foraminiferal assemblages between 500–2000 m depth off Hachinohe (northeast Japan). They reported that the *B. spissa* distribution is restricted to stations bathed by dysoxic waters between 760–1250 m depth. This species is reported from shallower than a water depth of 2319 m in the Japan Trench area (Fig. 10; Thompson, 1980), thus sediment reworking at shallower water depths by an episodic event and subsequent sedimentation of reworked particles may have occurred following the earthquake.

Although a few specimens of living foraminifera were found in the upper part of clusters Ib and IIa, these specimens were mainly composed of shallow infauna species. These living specimens might have been rapidly buried and isolated from an oxic condition by an episodic sedimentation event, and therefore the cytoplasm might not have decomposed. Above cluster IIa, Rose Bengal stained specimens increased to the upper part of the core; thus, the living assemblage in the upper part is regarded as the exact living assemblage.

Cluster II of the benthic foraminifera is equivalent to sediment facies III, the diatom ooze sediment deposited following the Fukushima-Daiichi Nuclear Power plant accident. The foraminiferal density drastically decreased in cluster I and many species of cluster I disappeared in subcluster IIa (9–7 cm depth), resulting in a decrease in species diversity. Living specimens of *Brizalina pacifica* and *Stainforthia* cf. *apertura* appeared in subcluster IIb (7–4 cm depth), then *S.* cf. *apertura*, *Nonionella* spp. (mainly composed of *N. globosa*), and *N. labradorica* became dominant in subcluster IIc (4–0 cm depth). Thus, the episodic deposition related to the earthquake resulted in a drastic faunal change in the pre-earthquake benthic foraminifera, and a new assemblage of low diversity was established. Similar low diversity assemblages influenced by the deposition of turbidities have been reported in upper continental slope and deep-sea canyons (Hess et al., 2005; Hess and Jorissen, 2009; Toyofuku et al., 2014; Bolliet et al., 2014). These studies showed that opportunistic taxa colonized and rapidly flourished after the disturbance of the seafloor. Kitazato et al. (2000) monitored the response of benthic foraminifera to seasonal inputs of organic matter from phytoplankton in Sagami Bay, Japan, and showed that the density of *Brizalina pacifica* (reported as *Bolivina pacifica* in their study) and *S. apertura* rapidly increased in response to the spring bloom at the ocean surface. A spring bloom of phytoplankton was observed between late March and early April 2011 off the Tohoku area, and depositions of phytodetritus aggregations were confirmed along the Japan Trench slope (Oguri et al., 2013). Thus, rapid deposition of phytoplankton may have enhanced the flourishing of these opportunistic taxa during the 17 months following the earthquake. *Nonionella globosa* is a shallow-to-deep infauna and *N. labradorica* is an intermediate-to-deep infauna (Fontanier et al., 2014). These species are putatively deposit feeders, and more competitive than others (Fontanier et al., 2014), thus these species also could flourish in the top of the core. Although the abundances were low, these species were also found in cores 2W-2011 and





2W-2012 as dominant species (Figs. 6 and 7). The vertical distribution pattern of increasing above 5 cm is similar among the three cores. Thus, faunal change may have already occurred 5 months after the earthquake. Hess et al. (2005) reported that the recovery of benthic foraminiferal assemblages from the impact of turbidite deposition was in the 1.5 year, and the community structure was in an early stage of recolonization. In our present study, some species belonging to cluster Ia (pre-earthquake) were found from the sediment surface (e.g., *Fursenkoina complanata*), suggesting an early recovery of original assemblages. However, it may take a much longer time to recover original assemblages because the disturbance at the 4W sites might have been more severe.

## 5. Conclusions

In this study, we examined the sediment characteristics and foraminiferal assemblages in the sediment cores collected from the landward slope of the Japan Trench around the hypocentral region 5 and 17 months after the 2011 Tohoku-oki earthquake (Mw 9.0) to document the impact of large-scale seafloor disturbance on a deep-sea benthic ecosystem.

[134]Cs from the Fukushima Daiichi Nuclear Power Plant accident was detected in three of the four examined cores. In particular, [134]Cs was detected in the top 0–10 cm of core 4W-2012, collected at the 4W site. The core was collected in August 2012, thus the sedimentation rate was estimated as 6 cm/year. This high sedimentation rate is probably related to the mass bloom of phytoplankton and the canyon topography where sediments are eroded/deposited.

We found a benthic foraminiferal faunal change to the deep-sea floor disturbance in the landward slope of the Japan Trench as follows.

The "original" assemblage, consisting of *Uvigerina senticosa*, *Chilostomella oolina*, and *Elphidium batialis*, was covered by sediment with a high diversity allochthonous assemblage related to event deposits following the earthquake. After the episodic deposition, the density drastically decreased and many species disappeared, resulting in a decrease in species diversity. Living specimens of opportunistic species, such as *Brizalina pacifica* and *Stainforthia* cf. *apertura*, and competitive species, such as *Nonionella* spp. and *N. labradorica*, gradually increased toward the sediment surface and became dominant in the top of the core. These faunal changes may have already occurred 5 months after the earthquake, and rapid deposition of phytoplankton may have enhanced the flourishing of these opportunistic taxa in the high sedimentation area. Further study will show the long-term recovery process of this deep-sea benthic ecosystem following mass disturbance of the seafloor.

## Acknowledgements

We would like to thank the crews and scientists of the *R/V Yokosuka* and the *HOV Shinkai 6500* of the Japan Agency for Marine–Earth Science and Technology (JAMSTEC) during the YK11-E06 and YK12-13 cruises. This study was financially supported by the research project Tohoku Ecosystem-Associated Marine Sciences from the Ministry of Education, Culture, Sports, Science, and Technology.

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



Fig. 1. Bathymetry of the studied area (A) and detailed sampling location of the sediment cores (B and C). The star symbol and open circles in figure 1-A indicate the epicenter of the 2011 off the Pacific coast of Tohoku earthquake and benthic foraminiferal samples used in Thompson (1980), respectively. Bathymetric map in figure 1-A was drawn by The Generic Mapping Tools (GMT) with bathymetric data from National Oceanic and Atmospheric Administration (NOAA).





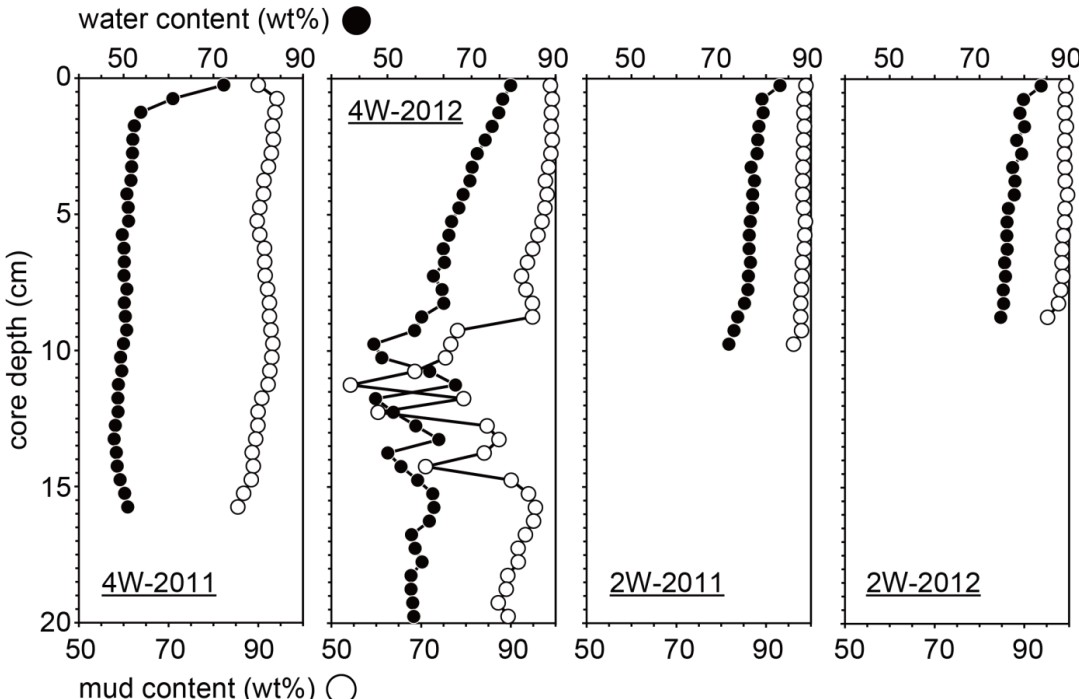

Fig. 2. Vertical profiles of water and mud content in sediments.





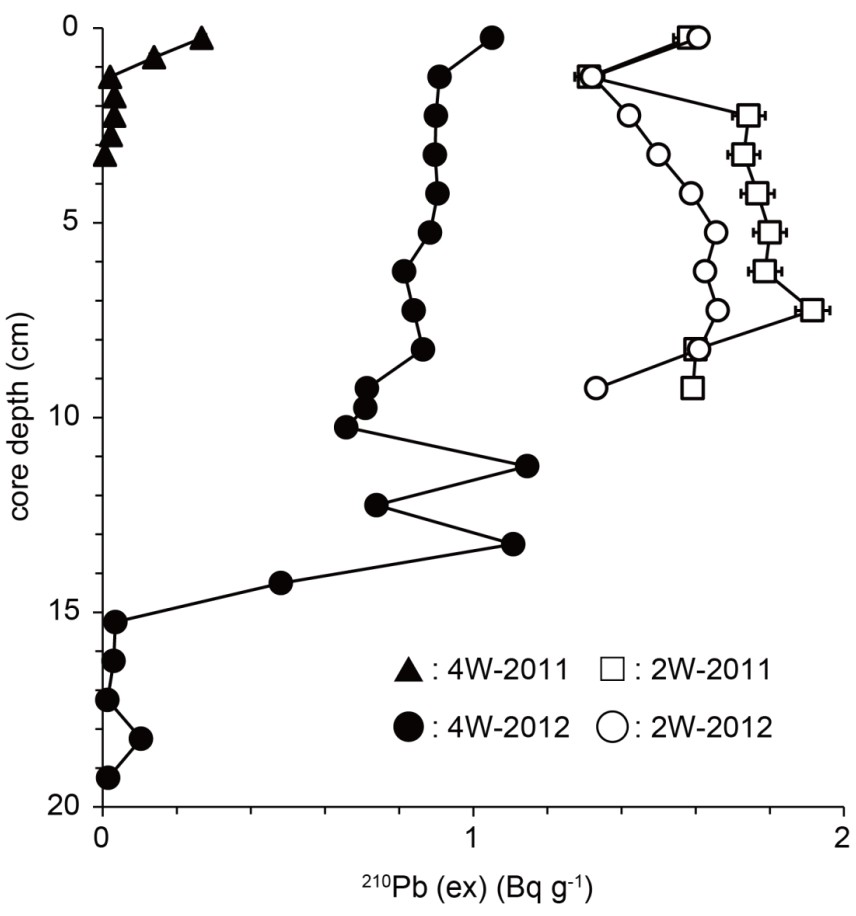

Fig. 3. Vertical profiles of the excess $^{210}$Pb in sediments. concentrations decay-corrected to sampling date.



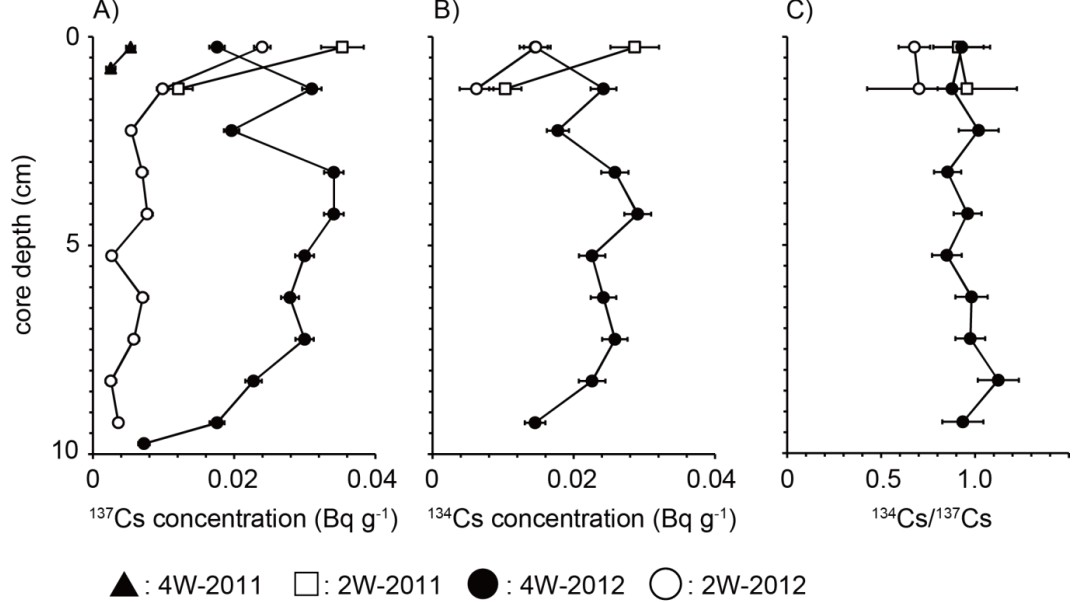

Fig. 4. Vertical profiles of A) $^{137}$Cs concentrations decay-corrected to sampling date; B) $^{134}$Cs concentrations decay-corrected to sampling date; C) $^{134}$Cs/$^{137}$Cs ratio decay-corrected to the FNPP1 accident.



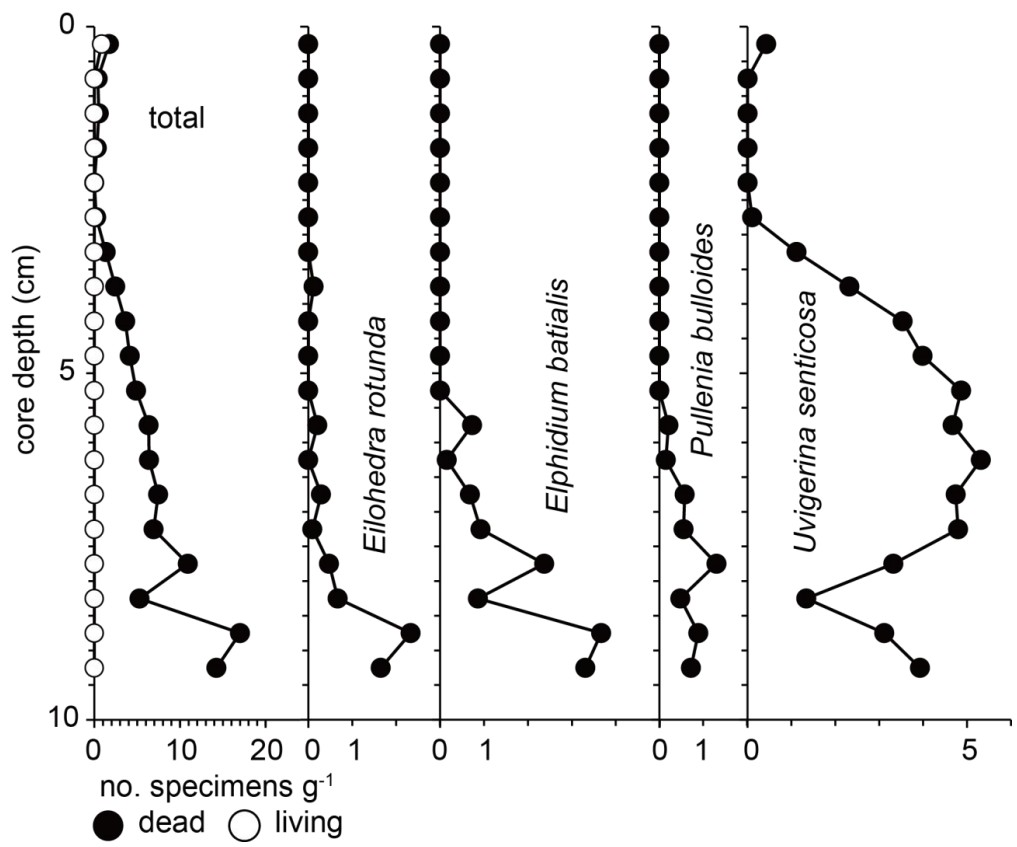

Fig. 5. Vertical distributions of absolute number of specimens per gram dry sediment of total foraminifera and dominant foraminiferal species in core 4W-2011.





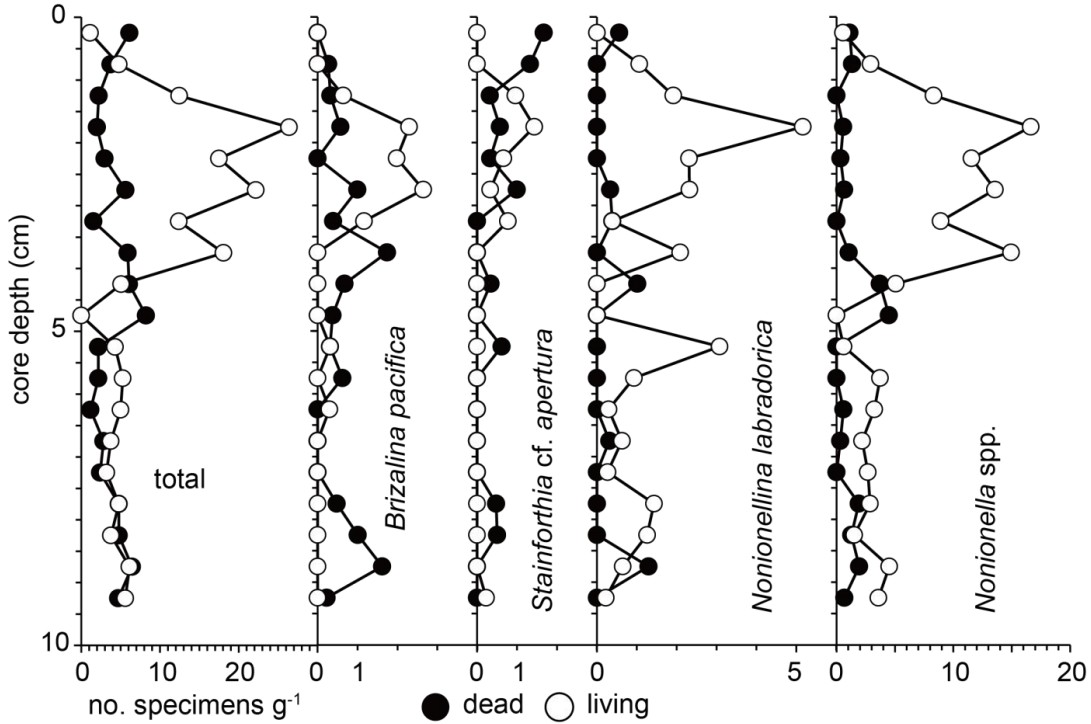

Fig. 6. Vertical distributions of absolute number of specimens per gram dry sediment of total foraminifera and dominant foraminiferal species in core 2W-2011. Open circles and filled circles indicate living and dead foraminifera, respectively.



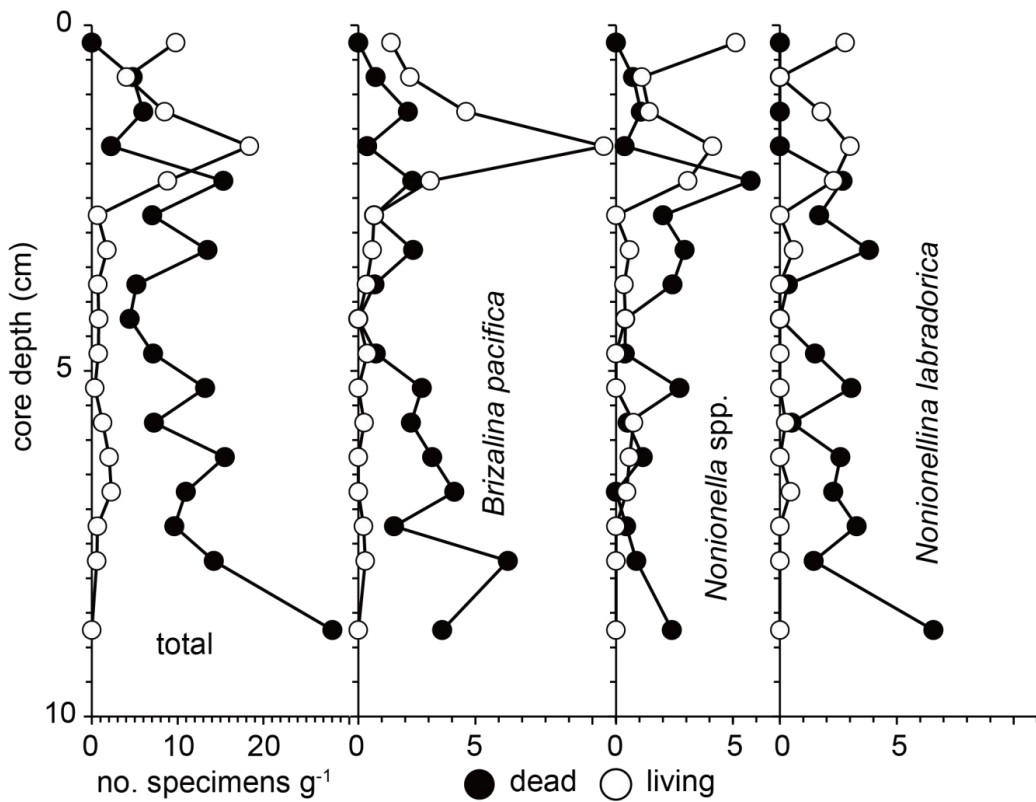

Fig. 7. Vertical distributions of absolute number of specimens per gram dry sediment of total foraminifera and dominant foraminiferal species in core 2W-2012. Open circles and filled circles indicate living and dead foraminifera, respectively.

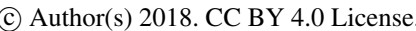





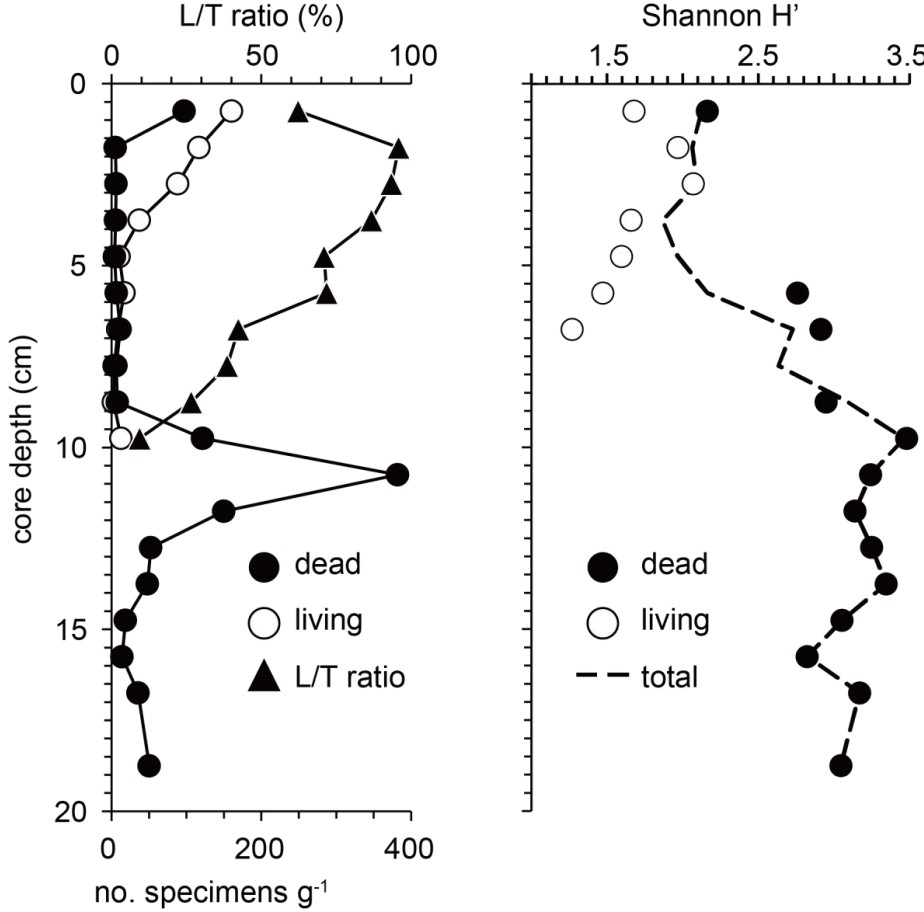

Fig. 8. Vertical profiles of dead and living foraminiferal density (ind. g⁻¹), and diversity index of dead, living, and total foraminiferal assemblage in core 4W-2012.



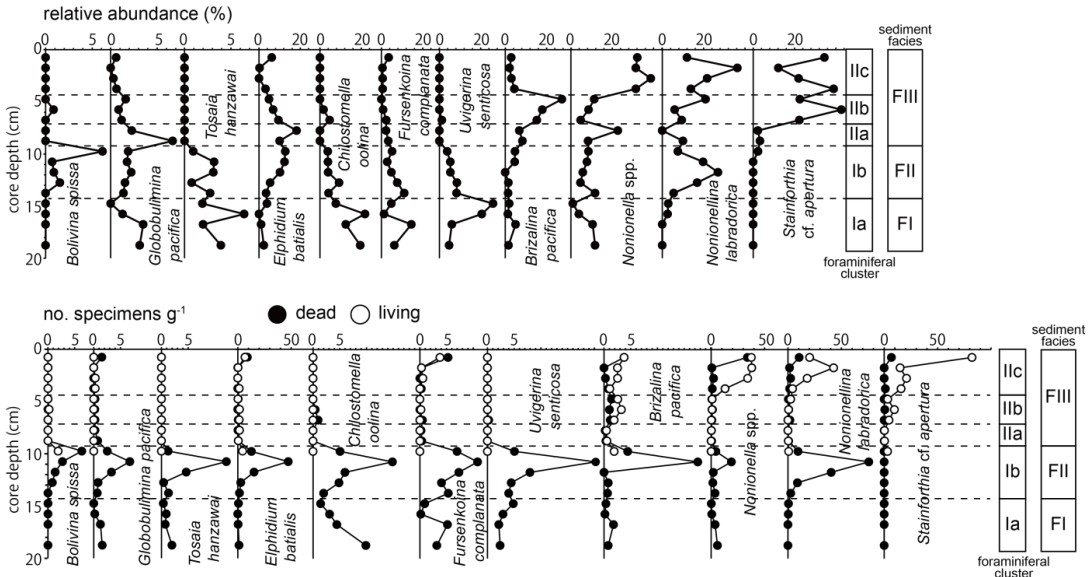

Fig. 9. Vertical distributions of relative and absolute number of specimens per gram dry sediment of dominant and characteristic foraminiferal species with Q-mode cluster grouping and sediment facies (FI, FII, and FIII) in core 4W-2012. Open circles and filled circles indicate living and dead foraminifera, respectively.



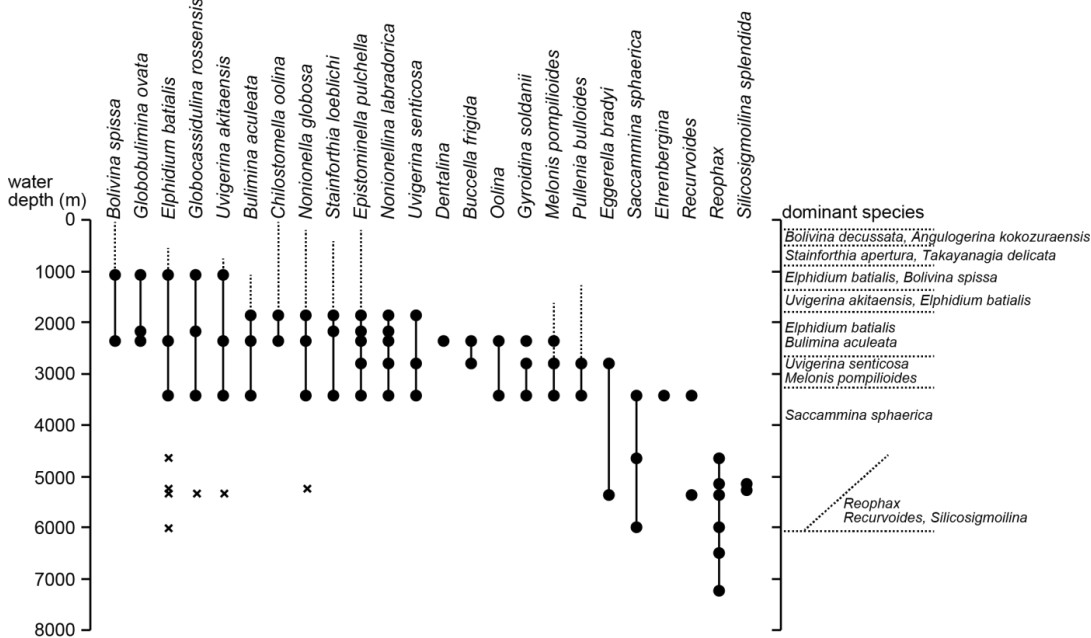

Fig. 10. Bathymetric distributions of benthic foraminiferal species in the Japan Trench area (modified after Thompson, 1980).

Table 1. Details of sampling sites of the sediment cores. *Station name used in Tsuji et al. (2013).

| Core ID | Station* | Cruise ID | Submersible dive number | Sampling date | Latitude | Longtitude | Water depth (m) |
|---------|----------|-----------|-------------------------|---------------|----------|------------|-----------------|
| 4W-2011 | 4W | YK11-E06 | 1257 | 5 Aug 2011 | 37°44.54'N | 143°17.06'E | 3566 |
| 2W-2011 | 2W | YK11-E06 | 1259 | 10 Aug 2011 | 38°39.28'N | 143°35.40'E | 3230 |
| 4W-2012 | 4W | YK12-13 | 1310 | 18 Aug 2012 | 37°44.24'N | 143°17.03'E | 3585 |
| 2W-2012 | 2W | YK12-13 | 1312 | 21 Aug 2012 | 38°39.34'N | 143°35.37'E | 3230 |

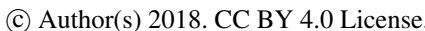



Table 2. γ-Spectrometry results of the sediment samples examined. Error indicates a standard deviation of counting statistics. *Concentrations decay-corrected based on sampling date. **Concentrations decay-corrected based on date of the FNPP1 accident. The "b.d.l." denotes "below detection limit".

| | mean depth (cm) | excess $^{210}$Pb* (Bq g$^{-1}$) | | | $^{137}$Cs* (Bq g$^{-1}$) | | | $^{134}$Cs * (Bq g$^{-1}$) | | | $^{134}$Cs /$^{137}$Cs** | | |
|---|---|---|---|---|---|---|---|---|---|---|---|---|---|
| 4W-2011 | 0.25 | 0.268 | ± | 0.011 | 0.005 | ± | 0.001 | b.d.l. | | | --- | | |
| | 0.75 | 0.139 | ± | 0.011 | 0.003 | ± | 0.001 | b.d.l. | | | --- | | |
| | 1.25 | 0.021 | ± | 0.006 | b.d.l. | | | b.d.l. | | | --- | | |
| | 1.75 | 0.032 | ± | 0.006 | b.d.l. | | | b.d.l. | | | --- | | |
| | 2.25 | 0.032 | ± | 0.007 | b.d.l. | | | b.d.l. | | | --- | | |
| | 2.75 | 0.023 | ± | 0.007 | b.d.l. | | | b.d.l. | | | --- | | |
| | 3.25 | 0.006 | ± | 0.007 | b.d.l. | | | b.d.l. | | | --- | | |
| 2W-2011 | 0.25 | 1.583 | ± | 0.042 | 0.035 | ± | 0.003 | 0.029 | ± | 0.003 | 0.91 | ± | 0.13 |
| | 1.25 | 1.313 | ± | 0.039 | 0.012 | ± | 0.002 | 0.010 | ± | 0.002 | 0.96 | ± | 0.27 |
| | 2.25 | 1.743 | ± | 0.044 | b.d.l. | | | b.d.l. | | | --- | | |
| | 3.25 | 1.730 | ± | 0.043 | b.d.l. | | | b.d.l. | | | --- | | |
| | 4.25 | 1.767 | ± | 0.045 | b.d.l. | | | b.d.l. | | | --- | | |
| | 5.25 | 1.801 | ± | 0.045 | b.d.l. | | | b.d.l. | | | --- | | |
| | 6.25 | 1.787 | ± | 0.045 | b.d.l. | | | b.d.l. | | | --- | | |
| | 7.25 | 1.916 | ± | 0.046 | b.d.l. | | | b.d.l. | | | --- | | |
| | 8.25 | 1.601 | ± | 0.023 | b.d.l. | | | b.d.l. | | | --- | | |
| | 9.25 | 1.593 | ± | 0.023 | b.d.l. | | | b.d.l. | | | --- | | |
| 4W-2012 | 0.25 | 1.098 | ± | 0.022 | 0.018 | ± | 0.001 | 0.015 | ± | 0.002 | 0.93 | ± | 0.15 |
| | 1.25 | 0.950 | ± | 0.019 | 0.031 | ± | 0.001 | 0.024 | ± | 0.002 | 0.88 | ± | 0.08 |
| | 2.25 | 0.940 | ± | 0.021 | 0.020 | ± | 0.001 | 0.018 | ± | 0.002 | 1.02 | ± | 0.11 |
| | 3.25 | 0.938 | ± | 0.021 | 0.034 | ± | 0.001 | 0.026 | ± | 0.002 | 0.85 | ± | 0.07 |
| | 4.25 | 0.945 | ± | 0.021 | 0.034 | ± | 0.001 | 0.029 | ± | 0.002 | 0.96 | ± | 0.07 |
| | 5.25 | 0.924 | ± | 0.021 | 0.030 | ± | 0.001 | 0.023 | ± | 0.002 | 0.85 | ± | 0.08 |
| | 6.25 | 0.851 | ± | 0.020 | 0.028 | ± | 0.001 | 0.024 | ± | 0.002 | 0.98 | ± | 0.09 |
| | 7.25 | 0.877 | ± | 0.020 | 0.030 | ± | 0.001 | 0.026 | ± | 0.002 | 0.97 | ± | 0.08 |
| | 8.25 | 0.904 | ± | 0.020 | 0.023 | ± | 0.001 | 0.023 | ± | 0.002 | 1.12 | ± | 0.11 |
| | 9.25 | 0.746 | ± | 0.019 | 0.018 | ± | 0.001 | 0.015 | ± | 0.001 | 0.94 | ± | 0.11 |
| | 9.75 | 0.741 | ± | 0.019 | 0.007 | ± | 0.001 | b.d.l. | | | --- | | |
| 2W-2012 | 0.25 | 1.609 | ± | 0.024 | 0.024 | ± | 0.001 | 0.015 | ± | 0.002 | 0.68 | ± | 0.08 |
| | 1.25 | 1.322 | ± | 0.022 | 0.010 | ± | 0.001 | 0.006 | ± | 0.002 | 0.70 | ± | 0.28 |
| | 2.25 | 1.421 | ± | 0.023 | 0.005 | ± | 0.001 | b.d.l. | | | --- | | |
| | 3.25 | 1.501 | ± | 0.023 | 0.007 | ± | 0.001 | b.d.l. | | | --- | | |
| | 4.25 | 1.589 | ± | 0.024 | 0.008 | ± | 0.001 | b.d.l. | | | --- | | |
| | 5.25 | 1.656 | ± | 0.024 | 0.003 | ± | 0.001 | b.d.l. | | | --- | | |
| | 6.25 | 1.626 | ± | 0.024 | 0.007 | ± | 0.001 | b.d.l. | | | --- | | |
| | 7.25 | 1.660 | ± | 0.024 | 0.006 | ± | 0.001 | b.d.l. | | | --- | | |
| | 8.25 | 1.610 | ± | 0.024 | 0.003 | ± | 0.001 | b.d.l. | | | --- | | |
| | 9.25 | 1.333 | ± | 0.022 | 0.004 | ± | 0.001 | b.d.l. | | | --- | | |

**Title for supplementary table**

**Table S1.** Foraminiferal species identified in this study.