# Peer review of "Impact of the 2011 off the Pacific coast of Tohoku earthquake on a deepsea benthic ecosystem: evidence from living and dead benthic foraminifera on the landward slope of the Japan Trench"

_Biogeosciences, 2018_

## Referee Comment (RC1) · Anonymous Referee #1 · 18 Jul 2018

I would like to congratulate the authors on their work. This was a very interesting manuscript and its always exciting to see the diversity of tsunami foraminiferal signatures in the sedimentary record explored, particularly from a deep-sea setting.

I think this manuscript needs moderate amendments before publication. My main concerns are largely cosmetic and relate to further detail (substantially more in some areas), clarification and explanation of various elements. I would also advise the use of more cautious language to describe your results as they're based on some relatively low sample sizes in some areas.

BGD questions: 1. Does the paper address relevant scientific questions within the scope of BG? Yes 2. Does the paper present novel concepts, ideas, tools, or data? Yes 3. Are substantial conclusions reached? Yes (with additional detail) 4. Are the scientific methods and assumptions valid and clearly outlined? Mostly 5. Are the results sufficient to support the interpretations and conclusions? Mostly 6. Is the description of experiments and calculations sufficiently complete and precise to allow their reproduction by fellow scientists (traceability of results)? Yes 7. Do the authors give proper credit to related work and clearly indicate their own new/original contribution? Yes 8. Does the title clearly reflect the contents of the paper? Yes but needs cutting back 9. Does the abstract provide a concise and complete summary? Yes 10. Is the overall presentation well structured and clear? Yes 11. Is the language fluent and precise? Needs work 12. Are mathematical formulae, symbols, abbreviations, and units correctly defined and used? Yes 13. Should any parts of the paper (text, formulae, figures, tables) be clarified, reduced, combined, or eliminated? Some clarification is needed in areas of the text. 14. Are the number and quality of references appropriate? Yes 15. Is the amount and quality of supplementary material appropriate? Further supplementary material would help this paper and future researchers.

Scientific questions: • Pg 3 Lines 11-13: Did you complete any sedimentological analyses or recordings of your cores? Any grainsize or stratigraphic record that you can link your foraminiferal assemblages to? Or sediment coloration (e.g. Munsell colours)? Perhaps an image of the core? This would be particularly useful in highlighting your allochthonous sediments and where you mention the deposition of diatomaceous ooze. • Pg 3 Line 17: What temperature did you oven dry the samples at? This can affect the preservation of agglutinated specimens. • Pg 3 Line 40: Why a 106 $\mu$m sieve for foraminiferal analyses? That sieve size is an usual gauge and you'd already sieved at 63 $\mu$m which is standard. Given the depth of your samples (I've found them as small as 30 $\mu$m from a nearby sample), a 106 $\mu$m sieve would have caused the loss of many, small heterotrophic species. • Pg 4 Section 3.1: Could you explain the significance of measuring mud and water content stability/lack of stability? • Pg 4

Section 3.3: Given the inherent difficulties in foram taxonomy and their potential benefit in being identified to species level in biogeographical studies, it would be good to have a plate of the identified species either as a figure or as a supplementary file/appendix to help future researchers. • Pg 4 Section 3.3: How many species of foraminifera both live and total were identified? • Pg 5 Section 3.3: I would like to see a figure that illustrates the clustering discussed, particularly showing the sub-clustering of the two main assemblages. This can be either as a figure in the main manuscript or as a supplement/appendix. Given the low raw numbers in some sections of core I'm not sure such extensive subdivision is necessary, and I think their subdivision confuses the story you're trying to tell. • Pg 6, section 4.2: You link your foram assemblages to 3 sediment facies that you never explained in your results. What facies? You cannot discuss these without establishing them in the first place. Just saying they exist and marking them on a figure is not enough. • Pg 6, Line 14: Why would diatom blooms accelerate 134Cs deposition? • Pg 6, Lines 19-21: You talk about diatoms and radiolarians in your discussion but they're not mentioned in your methodology or results. How have you quantified them and their significance? Why would diatom blooms accelerate 134Cs deposition? Why mention the radiolarians? • Pg 7, Line 10: "High species diversity" is relative given your small sample size, actual diversity values would be clearer here. • Pg 8, Section 5: Despite mentioning it in both your abstract and your conclusions nowhere do you report on your allochthonous foraminifera that are deposited between your pre-earthquake and post-earthquake/opportunistic form assemblage. Or did you mean for your downslope transported foraminifera to be your allochthonous assemblage? In any case, it is not clear in your results/discussion.

Technical comments: • The title's wording is very long, contains unnecessary detail for a title and is a bit awkward. I would recommend altering to something like "Impact of the 2011 Tohoku-oki Earthquake on the deep-sea benthos: evidence from foraminifera of the Japan Trench slope". Even "Deep-sea benthic foraminiferal evidence of the 2011 Tohoku-oki Earthquake impact on the Japan Trench" as you only really mention the significance of the downward slope location of your samples in one sentence. •

The English needs correction and tightening up. Many sentences are too long and should be subdivided and the grammar needs a lot of work. • "The 2011 off the Pacific coast of Tohoku earthquake" is a very long-winded name for an even that is well-known and greatly established both in the media and in scientific literature. It should just be referred to as the "2011 Tohoku-oki earthquake". • Figure 10: Why is there a dotted diagonal line above "Reophax Recurvoides, Silicosigmoilina" on the right-side Y-axis? • Pg 7, Line 3 (and elsewhere throughout the manuscript): You cannot start a sentence with an abbreviated species name, it needs to be written in full.

Please also note the supplement to this comment:
https://www.biogeosciences-discuss.net/bg-2018-237/bg-2018-237-RC1-supplement.pdf

---

## Referee Comment (RC2) · Anonymous Referee #2 · 19 Jul 2018

Tsujimoto et al, have tried to understand changes in benthic foraminifera following a powerful earthquake, by studying temporal changes in foraminiferal assemblage, in sediments deposited pre- and post-earthquake. They have identified disturbed sediments by using radionuclides. Similar studies have been carried out from the same region. The only novelty in this work is that the authors have taken a few cores from deeper region. I've several reservations regarding the methodology as well as the down-core variations in parameters, across the cores. 1. The title of the manuscript needs to be rephrased. Please see annotated manuscript for details. 2. The first

sentence of the introduction should be reworded. 3. Page 2, Line 11, please specify which fauna? The reference is also missing. 4. Page 2, Line 21, replace 'Their' with 'Foraminiferal'. 5. Can the authors explain the logic behind sampling during two consecutive years (2011 and 2012)? 6. Page 3, Line 14, the authors state that samples were stored at -80C for foraminiferal studies. Such a cold temperature will results in breakage of foraminiferal tests due to thawing. The breakage will thus significantly alter benthic foraminiferal assemblage. 7. The authors followed a strange methodology for foraminiferal analysis. To preserve and identify living benthic foraminifera, the sediments should be stained with rose-Bengal, immediately after collection. Unfortunately, the authors stored the sediments at -80C, oven dried it, sieved by using a 63 um sieve and THEN STAINED THE RESIDUE. I think, it is absolutely wrong. The living assemblage will be significantly underrepresented. 8. Staining the samples for just one day is insufficient. A minimum of couple of weeks of staining is widely recommended. 9. The authors chose a odd mesh size (106 um) to pick benthic foraminifera. The recommended mesh size for benthic foraminiferal study is 63 um. A few authors have also used >125 um fraction. The choice of these authors does not match with either of the widely used mesh size, thus making it difficult to compare their results with other studies. 10. Authors picked 200 or less foraminifera from each sample. Again, the recommended minimum number of specimens to be picked is 300. Therefore, I'm not sure whether the foraminiferal assemblage studied by the authors is a true representative of the natural assemblage. 11. It is not clear, how did the authors calculate foraminiferal density? Did you pick foraminifera from a known weight of sieved fraction? 12. Page 4, Line 4, authors state that they used only those samples that contained >30 individual, for statistical analysis. Does this mean that several samples contained as less as <30 individuals? It is too small a number to draw any meaningful inference from foraminiferal parameters. Please provide the number of specimens picked from each sample. 13. Three out of the four cores have a nearly same mud profile, suggesting no evidence of disturbance. 14. Can the authors provide the details of how far were the cores 4W-2011 and 4W-2012? Both these cores are very close and at nearly same

depth (just a difference of <20 m). Why was the 210Pb profile, so drastically different in so closely spaced cores? 15. The water depth of both the cores is nearly same, just 20 m difference. I do not agree with the authors that the slope is so different at these two locations, so as to bring such a big difference in geochemical parameters as well sedimentation rate. 16. Page 7, Line 19. Authors report living benthic foraminifera at much deeper depths and provide a strange explanation? Why this argument is not applicable for the other stained benthic foraminifera? How does the authors rule out the possibility that it is a autochthonous living benthic foraminifera? 17. Page 7, Line 36-37. The authors speculate the effect of productivity on benthic foraminifera. Please confirm it with the Corg in the sediments. In its present form, it is just a conjecture.

In view of the serious flaws in the methodology followed for foraminiferal study, I've strong reservations about this work.

Please also note the supplement to this comment:
https://www.biogeosciences-discuss.net/bg-2018-237/bg-2018-237-RC2-supplement.pdf

**Supplement:**

[revised manuscript text omitted]

---

## Author Comment (AC1) · 22 Aug 2018

Reviewer 1: I would like to congratulate the authors on their work. This was a very interesting manuscript and its always exciting to see the diversity of tsunami foraminiferal signatures in the sedimentary record explored, particularly from a deep-sea setting. I think this manuscript needs moderate amendments before publication. My main concerns are largely cosmetic and relate to further detail (substantially more in some areas), clarification and explanation of various elements. I would also advise the use of more cautious language to describe your results as they're based on some relatively

low sample sizes in some areas.

Response: Thank you very much for your positive and constructive comments on our manuscript. Reviewer's comments and suggestions were really helpful to improve our manuscript. Detailed responses to the comments are below. Added or improved sentences are written in red-letters in the manuscript.

Scientific questions: Pg 3 Lines 11-13: Did you complete any sedimentological analyses or recordings of your cores? Any grainsize or stratigraphic record that you can link your foraminiferal assemblages to? Or sediment coloration (e.g. Munsell colours)? Perhaps an image of the core? This would be particularly useful in highlighting your allochthonous sediments and where you mention the deposition of diatomaceous ooze.

Response: We performed mud contents analysis on our core as presented in the original manuscript, but we did not perform other sedimentological analyses of the analyzed cores. However, we collected other cores from same sites for detailed sedimentological analyses. In the revised manuscript, we added X-ray CT data as supplementary material (Figure S4) to provide more robust background of our findings on foraminiferal faunal changes. The image of 4W site-2012 clearly indicates event deposit comparable with our analyzed core. Laminated layer between ca. 11 to 5 cm depth in the image of 4W site-2012 is interpreted as turbidite. This turbidite layer coincides with sediment facies FII and foraminiferal cluster Ib in our core 4W-2012. We added this description in the text.

Pg 3 Line 17: What temperature did you oven dry the samples at? This can affect the preservation of agglutinated specimens.

Response: We oven-dried at 50 °C. We often use this temperature and confirm that this temperature does not affect the preservation of agglutinated specimens. We added the temperature in the text.

Pg 3 Line 40: Why a 106 $\mu$m sieve for foraminiferal analyses? That sieve size is an

usual gauge and you'd already sieved at 63 $\mu$m which is standard. Given the depth of your samples (I've found them as small as 30 $\mu$m from a nearby sample), a 106 $\mu$m sieve would have caused the loss of many, small heterotrophic species.

Response: This size fraction follows our conventional procedure by Nomura (1995), Tsujimoto et al. (2013), and Takata et al. (2015), which studied deep sea benthic foraminifera. We added these references in the text. We aware that most ecological studies used either 63 $\mu$m or 125 $\mu$m sieves, and the differences in sieve size in our studies hampers direct comparison with such ecological or environmental monitoring studies. We noted these methodological limitations in our revised manuscript. However, even though there are such limitations if we try to compare with other studies, we believe that the temporal and spatial trends found in our study based on consistent taxonomical works and worth to publish with such notes.

Pg 4 Section 3.1: Could you explain the significance of measuring mud and water content stability/lack of stability?

Response: Mud content varies in accordance with the changes in depositional condition. Generally, mud content becomes low under high-energy condition such as turbidites, and water content varies in accordance with the changes in mud content and sediment compaction. Thus, Mud and water content in sediment core become indicators for sedimentary environment. We added this explanation in the text.

Pg 4 Section 3.3: Given the inherent difficulties in foram taxonomy and their potential benefit in being identified to species level in biogeographical studies, it would be good to have a plate of the identified species either as a figure or as a supplementary file/appendix to help future researchers.

Response: We will add SEM images of dominant and characteristic species as a supplementary file.

Pg 4 Section 3.3: How many species of foraminifera both live and total were identified?

[Figure]

Response: Full faunal list was available in the supplementary Table S1, however, for better understandings for readers, we added number of specimens and number of species in Table S1.

Pg 5 Section 3.3: I would like to see a figure that illustrates the clustering discussed, particularly showing the sub-clustering of the two main assemblages. This can be either as a figure in the main manuscript or as a supplement/appendix. Given the low raw numbers in some sections of core I'm not sure such extensive subdivision is necessary, and I think their subdivision confuses the story you're trying to tell.

Response: We re-examined Q-mode cluster analysis based on the samples containing more than 50 individuals, and we added the result as a supplementary file (Figure S2). Although sub-cluster IIa was composed of one sample, the result of others did not change. The boundary of sub-cluster II-a and II-b correspond to the boundary of sediment facies, so we think sub-cluster is useful for discussion.

Pg 6, section 4.2: You link your foram assemblages to 3 sediment facies that you never explained in your results. What facies? You cannot discuss these without establishing them in the first place. Just saying they exist and marking them on a figure is not enough.

Response: We discussed 3 sediment facies based on radionuclides distribution in the last paragraph of section 4.1. Please see the text. We changed the name of sediment facies to FI, FII, and FIII in order to avoid confusion with foraminiferal cluster Ia, Ib, IIa, IIb, and IIc.

Pg 6, Line 14: Why would diatom blooms accelerate 134Cs deposition?

Response: Otosaka et al. (2014) studied time-series sinking particles from August 2011 to June 2013 at about 100 km east of the FNPP1. They reported that the production of diatoms and subsequent sinking of biogenic particles caused the increase in total mass flux of sinking particles from April to June. They concluded that adsorption or incorporation of radiocesium onto particles in the surface water and following rapid sinking of particles are considered as primary mechanisms of accumulation of radiocesium on the seafloor. We added these descriptions in the text with reference.

Pg 6, Lines 19-21: You talk about diatoms and radiolarians in your discussion but they're not mentioned in your methodology or results. How have you quantified them and their significance? Why would diatom blooms accelerate 134Cs deposition? Why mention the radiolarians?

Response: We observed the siliceous biogenic particles (diatoms and radiolarians) of the >106-$\mu$m fraction under stereoscopic microscope while picking benthic foraminifera. We added this explanation in methods. Although we performed only qualitative observation, the lower part of core 4W-2012 was rich in radiolarians and the upper part (above ca. 9 cm depth) is rich in diatoms. This observation, together with high TOC data added in the supplement, support the diatom bloom and its deposition after the earthquake. We added the photographs of the >106-$\mu$m fraction at representative depths (4.5-5.0 cm and 15.5-16.0 cm) as supplementary figure (Figure S5). Please also see above mentioned response about diatom blooms and 134Cs deposition.

Pg 7, Line 10: "High species diversity" is relative given your small sample size, actual diversity values would be clearer here.

Response: We recalculated diversity index based on the samples containing more than 50 individuals. We also determined the rarefaction diversity E (S100) (= the expected number of species in samples rarefied to 100 individuals) in addition to the Shannon Index (H) for total assemblage of core 4W-2012 (Fig. 8). We deleted the Shannon Index (H) for living and dead assemblages because of its small sample size. New diversity indices indicate relatively high species diversity in cluster Ib.

Pg 8, Section 5: Despite mentioning it in both your abstract and your conclusions nowhere do you report on your allochthonous foraminifera that are deposited between

your pre-earthquake and post-earthquake/opportunistic form assemblage. Or did you mean for your downslope transported foraminifera to be your allochthonous assemblage? In any case, it is not clear in your results/discussion.

Response: We discussed the possibility of reworking in section 4.2 by the appearance of Bolivina spissa, which is reported from shallower than a water depth of 2319 m in the Japan Trench area. We rephrased "allochthonous" in abstract and conclusions to "reworked".

Technical comments: The title's wording is very long, contains unnecessary detail for a title and is a bit awkward. I would recommend altering to something like "Impact of the 2011 Tohoku-oki Earthquake on the deep-sea benthos: evidence from foraminifera of the Japan Trench slope". Even "Deep-sea benthic foraminiferal evidence of the 2011 Tohoku-oki Earthquake impact on the Japan Trench" as you only really mention the significance of the downward slope location of your samples in one sentence.

Response: We changed title based on reviewer's suggestion as "Impact of the 2011 Tohoku-oki Earthquake on the deep-sea benthos: Evidence from foraminifera of the Japan Trench slope", and mentioned official earthquake name in the introduction

The English needs correction and tightening up. Many sentences are too long and should be subdivided and the grammar needs a lot of work.

Response: The original manuscript was edited by a native English speaker using a commercial service. We revised entire text in particular long sentences, and will have additional native English checks (probably by a different company) by an experienced editor whose first language is English and who is specialized in the editing of papers written by scientists whose native language is not English.

"The 2011 off the Pacific coast of Tohoku earthquake" is a very long-winded name for an even that is well-known and greatly established both in the media and in scientific literature. It should just be referred to as the "2011 Tohoku-oki earthquake".

Response: Because the official name of the earthquake given by Japan Meteorological Agency is "the 2011 off the Pacific coast of Tohoku earthquake", we used that also in the title, however, we agree that we do not need to use this as a title. We rephrase "2011 off the Pacific coast of Tohoku earthquake" to "2011 Tohoku-oki earthquake" in Introduction, so we used "2011 Tohoku-oki earthquake" in other part as reviewer's suggestion.

Figure 10: Why is there a dotted diagonal line above "Reophax Recurvoides, Silicosigmoilina" on the right-side Y-axis?

Response: This figure is quotation from Thompson (1980). The dotted diagonal line indicates differences between west side and east side across the Japan Trench. We added this explanation in the figure caption.

Pg 7, Line 3 (and elsewhere throughout the manuscript): You cannot start a sentence with an abbreviated species name, it needs to be written in full.

Response: We corrected as the reviewer 1 suggested.

Please also note the supplement to this comment:
https://www.biogeosciences-discuss.net/bg-2018-237/bg-2018-237-AC1-supplement.zip

---

## Author Comment (AC2) · 22 Aug 2018

Tsujimoto et al, have tried to understand changes in benthic foraminifera following a powerful earthquake, by studying temporal changes in foraminiferal assemblage, in sediments deposited pre- and post-earthquake. They have identified disturbed sediments by using radionuclides. Similar studies have been carried out from the same region. The only novelty in this work is that the authors have taken a few cores from deeper region. I've several reservations regarding the methodology as well as the down-core variations in parameters, across the cores.

[Figure]

Response: We would like to thank anonymous referee 2 for critical and constructive comments. Reviewer's comments were really helpful to improve our manuscript. Detailed responses to the review comments are below. All technical comments on supplementary file were corrected or incorporated. Added or improved sentences are written in red-letters in the manuscript.

1. The title of the manuscript needs to be rephrased. Please see annotated manuscript for details.

Response: We changed title based on the suggestions of referees 1and 2 as "Impact of the 2011 Tohoku-oki Earthquake on the deep-sea benthos: Evidence from foraminifera of the Japan Trench slope".

2. The first sentence of the introduction should be reworded.

Response: We changed the first sentence of the introduction.

3. Page 2, Line 11, please specify which fauna? The reference is also missing.

Response: We added the species name (Psammosphaera spp.) and reference (Toyofuku et al., 2014).

4. Page 2, Line 21, replace 'Their'with 'Foraminiferal'.

Response: We corrected.

5. Can the authors explain the logic behind sampling during two consecutive years (2011 and 2012)?

Response: We added short description on sampling strategy in line 4 to 6 of page 3 and also limitations on consistent samplings between 2011 and 2012 in line 7 to 11 of page 3.

6. Page 3, Line 14, the authors state that samples were stored at -80C for foraminiferal studies. Such a cold temperature will results in breakage of foraminiferal tests due to

thawing. The breakage will thus significantly alter benthic foraminiferal assemblage.

Response: In case biological samples, it is preferable to freeze at lower temperature such as -80dC or even lower, because freezing at -20dC or -30dC can cause cell destruction (e.g. McHatton et al. 1996, Appl. Environ. Microbiol. 62:954–958.) due to water crystal formation. We thus think that -80dC preservation may reduce these physical destructions even for foraminiferal test. However, we agree that freezing and staining with rose-Bengal afterward is not common protocol in foraminiferal study and may have biased our results. We noted potential effects of freezing in the discussion.

7. The authors followed a strange methodology for foraminiferal analysis. To preserve and identify living benthic foraminifera, the sediments should be stained with rose-Bengal, immediately after collection. Unfortunately, the authors stored the sediments at -80C, oven dried it, sieved by using a 63 um sieve and THEN STAINED THE RESIDUE. I think, it is absolutely wrong. The living assemblage will be significantly underrepresented. 8. Staining the samples for just one day is insufficient. A minimum of couple of weeks of staining is widely recommended.

Response: In this study, we followed the traditional procedure for rose-Bengal staining method by Takayanagi (1978, Manual of Microfossil Studies) and subsequent studies mainly by Japanese researchers (e.g., Tsujimoto et al., 2006; Nomura and Kawano, 2011). Also, we did NOT oven-dry the sediment before staining; we revised the text to make it clear. We understand the importance of the standard staining method as the reviewer pointed out, especially for the benthic foraminiferal monitoring studies (e.g., Schönfeld et al., 2012). However, although the living assemblage may be underrepresented, we mainly discuss temporal faunal changes based on total (i.e. living and dead) foraminifera. Furthermore, in the revised manuscript, we discuss more based on total fauna, not by "living" fauna which may represent underestimated numbers. We noted these methodological limitations in the discussion. Although these limitations on methodologies to study on "living" fauna, , temporal trend in total foraminiferal assemblage indicates the impact of large-scale seafloor disturbance on deep-sea benthic

ecosystem together with concurrent sedimentological analyses, thus we believe these results are worth to be published in this journal.

9. The authors chose a odd mesh size (106 um) to pick benthic foraminifera. The recommended mesh size for benthic foraminiferal study is 63 um. A few authors have also used >125 um fraction. The choice of these authors does not match with either of the widely used mesh size, thus making it difficult to compare their results with other studies.

Response: This size fraction follows our conventional procedure by Nomura (1995), Tsujimoto et al. (2013), and Takata et al. (2015), which studied deep sea benthic foraminifera. We added these references in the text. We aware that most ecological studies used either 63 $\mu$m or 125$\mu$m sieves, and the differences in sieve size in our studies hampers direct comparison with such ecological or environmental monitoring studies. We noted these methodological limitations in our revised manuscript. However, even though there are such limitations when comparing with other studies, we believe that the temporal and spatial trends found in our study based on consistent taxonomical works are worth to publish with such notes.

10. Authors picked 200 or less foraminifera from each sample. Again, the recommended minimum number of specimens to be picked is 300. Therefore, I'm not sure whether the foraminiferal assemblage studied by the authors is a true representative of the natural assemblage.

Response: We had picked 300 specimens in some samples, but some samples contained individual numbers less than 200 individuals even if we analyses entire sediment samples ($\sim$12 ml). Full faunal list was available in the supplementary Table S1, however, for better understandings for readers, we added number of specimens and number of species in Table S1.

11. It is not clear, how did the authors calculate foraminiferal density? Did you pick foraminifera from a known weight of sieved fraction?

Response: We measured the wet weights of sliced subsamples before wet-sieving through a 63-$\mu$m sieve. We calculated the dried weights of sliced subsamples from the water content obtained from the process of water content analysis (see section 2.2). The foraminiferal density (i.e. number of foraminifera per gram of dry sediment) was calculated from the dried weights of sliced subsamples. We added this explanation in Method.

12. Page 4, Line 4, authors state that they used only those samples that contained >30 individual, for statistical analysis. Does this mean that several samples contained as less as <30 individuals? It is too small a number to draw any meaningful inference from foraminiferal parameters. Please provide the number of specimens picked from each sample.

Response: We added the number of specimens of each sample in supplementary Table S1. We performed statistical analysis only for total assemblage of core 4W-2012 because the others did not contain sufficient specimens. We recalculated Q-mode cluster analysis and diversity index based on the samples containing more than 50 individuals. We also determined the rarefaction diversity E (S100) (= the expected number of species in samples rarefied to 100 individuals) in addition to the Shannon Index (H) for total assemblage of core 4W-2012. The results from additional analyses showed no obvious change from the previous analyses.

13. Three out of the four cores have a nearly same mud profile, suggesting no evidence of disturbance.

Response: As the reviewer suggested, evidence of disturbance is not obvious from mud content of core 2W-2011 and 2W-2012, but detection of 134Cs in the top 1.5 cm of the cores indicates that this interval was deposited after the FNPP1 accident. Moreover, relative low concentrations of 210Pb in the top 1.5 cm of the cores 2W may indicate that deposition of older sediments derived from seafloor disturbance. The rapid decrease of water content and the very low 210Pb concentrations in core 4W-

2011 suggest that the surface sediments at this site were eroded prior to the sampling.

14. Can the authors provide the details of how far were the cores 4W-2011 and 4W-2012? Both these cores are very close and at nearly same depth (just a difference of <20 m). Why was the 210Pb profile, so drastically different in so closely spaced cores?

Response: We added bathymetric profile at site 4W as supplementary figure (Figure S3). Although cores 4W-2011 and 4W-2012 were collected in the same area, which are ∼550 m away each other, slight locational changes in the canyon should result in different sedimentation environments. Our new supplementary figure S3 showing the bathymetric profile along the two sites clearly shows that 4W-2011 was collected from a steeper slope (∼4°) while 4W-2012 was collected from a gentle slope (∼1°) of the same canyon (Fig. S4). Sedimentation rates can vary greatly even at local scale, in particular at such topographic and hydrographic heterogeneity environments. A flow such as a turbidity current was accelerated by an increase in slope gradient and eroded the surface sediment near 4W-2011. The flow was decelerated by a decrease in the slope gradient and deposited the new turbidite near 4W-2012. As the consequence, the lithology and profiles of radionuclides between the two cores are quite different. We added these description in the text.

15. The water depth of both the cores is nearly same, just 20 m difference. I do not agree with the authors that the slope is so different at these two locations, so as to bring such a big difference in geochemical parameters as well sedimentation rate.

Response: As mentioned above (reply to comment 14), bottom topography differs largely between 2011 and 2012 sites, thus the geochemical parameters and sedimentation rates. Indeed, we originally thought these two cores are collected from comparable environments to track foraminiferal faunal change between 2011 and 2012, because these sites are almost same water depths and located at same canyon axis. However, detailed bathymetric survey and sedimentological analyses revealed that such local topographic change can cause great differences in sedimentation environments. We believe this is another important result from our study.

16. Page 7, Line 19. Authors report living benthic foraminifera at much deeper depths and provide a strange explanation? Why this argument is not applicable for the other stained benthic foraminifera? How does the authors rule out the possibility that it is a autochthonous living benthic foraminifera?

Response: As the reviewer pointed out, due to the limitation on staining methodologies different from standard protocols, we deleted these speculations from the text.

17. Page 7, Line 36-37. The authors speculate the effect of productivity on benthic foraminifera. Please confirm it with the Corg in the sediments. In its present form, it is just a conjecture.

Response: We added the TOC value of the 4W-2012 as supplementary table (Table S2), which represents very high TOC concentrations (∼4%) down to 10 cm depth. This TOC concentrations are very high given the water depth is 3600m, strongly support the rapid sedimentation of fresh organic matters. We added these data and descriptions in the text.

Comments in supplement file

Page 1, Line 25. Not clear, please rephrase.

Response: We rephrased as "After the disappearance of many benthic foraminiferal species".

Page 1, Line 29. Please specify, what you mean by faunal change.

Response: We specified as "the opportunistic species".

Page 3, Line 2. Not clear.... did you collect a couple of cores at each station? If so, why?

Response: We collected single core from at each site, from each sampling year, making four cores in total (2 sampling periods at 2 sites). We made these explanation clearer.

Page 2, Line 11; Page 3, Line 2; Page 3, Line 39 - Page 4, Line 5; Page 4, Line 24 - Line 28; Page 5, Line 36 - Line 38; Page 7, Line 19; Page 7, Line 35 – Line 37

Please see above mentioned responses.

Please also note the supplement to this comment:
https://www.biogeosciences-discuss.net/bg-2018-237/bg-2018-237-AC2-supplement.zip